# Seepage Characteristics of a Single Ascending Relief Well Dewatering an Overlying Aquifer

**Wenxue Wang** [1],*[ID]**, Boris Faybishenko** [2],*, **Tong Jiang** [1], **Jinyu Dong** [1] **and Yang Li** [3]

1   Henan Province Key Laboratory of Rock and Soil Mechanics and Structural Engineering, North China University of Water Resources and Electric Power, 136 Jinshui East Rd, Zhengzhou 450045, China; jiangtong@ncwu.edu.cn (T.J.); dongjinyu@ncwu.edu.cn (J.D.)
2   Energy Geosciences Division, Earth and Environmental Sciences Area, Lawrence Berkeley National Laboratory, University of California, Berkeley, CA 94720, USA
3   Hydrology and Water Resources Bureau of Henan Province, Zhengzhou 450003, China; laymanlee13@163.com
*   Correspondence: wangwenxue@ncwu.edu.cn (W.W.); bafaybishenko@lbl.gov (B.F.); Tel.: +86-15905211945 (W.W.)

**Abstract:** The application of groundwater relief, i.e., dewatering, ascending wells, drilled upward from the mining tunnel into the overlying aquifer, is common in underground mining engineering. In this study, the seepage characteristics of single ascending partially and fully penetrating relief wells are investigated using a series of laboratory sand-tank experiments and numerical simulations. The seepage characteristics of ascending wells dewatering an overlying aquifer are different from those of conventional pumping wells descending from the ground surface into the underlying aquifer, because of the pronounced influence of the seepage face boundary condition along the seepage boundary of the ascending dewatering well. The seepage face of the ascending well is formed as the well casing remains open and water is discharged under the action of gravity through the well casing. The results of laboratory sand-tank experiments and modeling show that when the degree of penetration of an ascending relief well does not exceed a critical value, the effect of the seepage face cannot be ignored. In particular, the seepage flux increases as the degree of penetration increases following an exponential function, and the relationship between the seepage flux and the well radius can be described using a power law function. The results of numerical simulations are used to develop a series of type curves to evaluate the effects of the critical degree of penetration for different well radii and different aquifer water levels. Modified versions of the Dupuit and Dupuit–Thiem formulae for a single ascending partially well for the degree of penetration less than the critical one for the unconfined, confined, and confined-unconfined aquifers are developed.

**Keywords:** ascending relief well; groundwater; seepage; sand-tank; modeling; Dupuit formula; Dupuit-Thiem formula

## 1. Introduction

Construction of many underground facilities and tunnels, deep foundation pits, and underground coal mines, which may be affected by underground water intrusion, require dewatering of an overlying aquifers [1–11]. A common method to reduce the groundwater level to prevent inundation of underground excavations is the use of water pumping wells, which are drilled from the land surface into the aquifer [5,7,8]. Another method applied in the mining industry for dewatering and depressurization of overlying aquifers is to drill dewatering wells from the underground mines' tunnels upward into the overlying aquifer [1,11–19].

In this paper, the dewatering wells drilled upward into the overlying aquifer are called "ascending relief wells" (ARWs), and they are used to partially or completely dewater an aquifer above an active or abandoned underground mine. The ARWs can also be used in combination with pumping wells. For example, a comprehensive drainage from pumping wells and the panels was used in the 8th mining area of Taiping Coal Mine, China [17]. The ascending wells were drilled below the mining face to test hydrological parameters in many underground coal mines [11,15]. The drainage water discharged under gravity from the overlying aquifer through ARWs is collected in the underground tunnel and then pumped out through the drainage system. Schematic diagrams of the partially penetrating ARWs in an unconfined aquifer, a confined aquifer, and a confined-unconfined aquifer are shown in Figure 1.

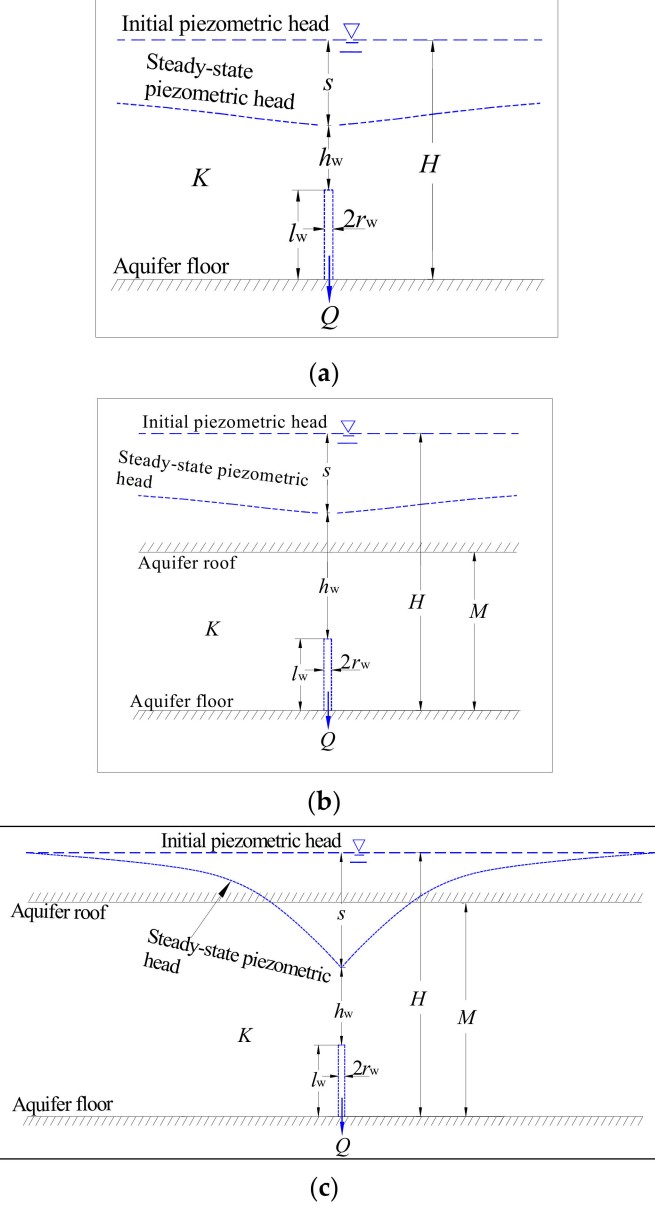

**Figure 1.** Schematic diagrams of a partially penetrating ascending relief well Ascending Relief Wells (ARW) in (**a**) an unconfined aquifer, (**b**) a confined aquifer, and (**c**) a confined-unconfined aquifer. On the figures, $s$ is the drawdown of water table, $h_w$ is the height of the water table above the well, $l_w$ is well screen length, $r_w$ is well radius, $H$ is the initial piezometric head, $M$ is the aquifer thickness, and $K$ is hydraulic conductivity.

Contrary to the conventional pumping wells, the casing of the ARW remains open to the ambient air with no standing water. The seepage characteristics of the ARWs are different from those of conventional pumping wells, because of downward water flow by the action of gravity into the underground tunnel. The physics of seepage through the walls of the ARWs is generally similar to that of seepage through the walls of the open tunnels. In particular, because of the formation of the seepage face at the tunnel walls [20–24], boundary conditions for numerical simulations of seepage into the tunnels are based on assigning the atmospheric pressure at the tunnel walls and a constant hydraulic head boundary condition along the tunnel perimeter [25–28]. The literature review shows that there is no generalized theory to assess the performance of ARWs [11,29,30], and determination of seepage characteristics of ARWs is based on using methods developed for conventional pumping wells [4,31–39].

The overall objectives of this paper are (1) to study the evolution of the seepage characteristics of single ascending partially and fully penetrating relief wells, and (2) to determine the critical degree of penetration of a partially penetrating ARW needed to obtain a maximum outflow under different far-field water level boundary conditions, (3) to assess the applicability and to modify the Dupuit and Dupuit–Thiem formulae for a single ascending partially well, in an unconfined, homogeneous and isotropic aquifer. The Dupuit and Dupuit–Thiem formulae were considered as a basis to calculate seepage fluxes of the ARW, and laboratory sand-tank experiments and numerical simulations were also conducted.

The following simplifying assumptions were taken into consideration in this study: (1) the aquifer is homogeneous, isotropic, and with no regional flow; (2) there is no water leakage through the top and bottom of the aquifer, i.e., through the underlying and overlying aquicludes of the confined aquifer; (3) the ARW is vertical with a screen along its entire length of the ARW penetration into the overlying aquifer, and no water column inside the ARW, allowing for free drainage (under the gravity), through the open well bottom, into the underlying tunnel, and there is no enforced pumping; (4) the atmospheric pressure boundary condition at the well-aquifer interface; and (5) groundwater flow in the aquifer toward the well is axisymmetric, and can be described by the Darcy law.

## 2. Methods of Analytical, Experimental, and Modeling Investigations

### 2.1. Dupuit and Dupuit–Thiem Formulae

Analysis of pumping wells drilled from the surface, in general, relies on the Dupuit assumption of a constant hydraulic head along the vertical profile of the aquifer. This assumption makes possible to use a depth-integrated equation for the flux evaluation into the well. Based on the original analytical solution for unconfined flow by Dupuit [40], Thiem [41] was likely the first to estimate aquifer parameter from pumping tests in a confined aquifer. Boulton [42] developed the first transient well test solution to analyze the unconfined aquifer. Streltsova [43] proposed the solution to consider unsteady radial flow in an unconfined aquifer. Neuman [44,45] developed solutions considering both confined storage and delayed yield from the unconfined aquifer. A variety of new analytical approaches were applied to deal with well hydraulics models involving mixed and complicated boundaries, which gained more accurate solutions for particular conditions [7,36,39,46–51]. Mishra and Kuhlman [52] have recently summarized the application of the Dupuit theory for an unconfined aquifer from Dupuit to the present. Charnyi [53] showed that the Dupuit formula for an unconfined aquifer is valid not only for a hydraulic approximation, i.e., assuming that the groundwater velocity is independent of the height above the aquitard, but also for rigorous hydrodynamic calculations. Although, the Dupuit formula is not accurate for calculations of the phreatic line for $r < H$, it provides accurate calculations of the seepage flux [53]. Shercliff [54] conducted one—and two-dimensional modeling of steady seepage flow in unconfined aquifers, and provided a proof of Charnyi's result that one—and two-dimensional theory yield the same value for the flow rate in a horizontal aquifer or porous bed between vertical ends, and showed the extent to which it can be generalized to non-uniform or anisotropic media.

The forms of the Dupuit and Dupuit–Thiem formulae are summarized in Table 1, and were used in this paper as a basis to estimate seepage fluxes into ARWs, and were then modified based on the results of laboratory sand-tank experiments and modeling (see Section 3).

**Table 1.** Formulae for steady-state seepage flux in unconfined, confined, and confined-unconfined aquifers.

| Types of Aquifers | Seepage Flux ($Q$) |
|:---:|:---:|
| Unconfined (Dupuit) | $Q = 1.366 \dfrac{K(2H-s)s}{\lg \dfrac{R}{r_w}}$ |
| Confined (Dupuit–Thiem) | $Q = 2.73 \dfrac{KMs}{\lg \dfrac{R}{r_w}}$ |
| Confined-Unconfined (Dupuit–Thiem) | $Q = 1.366 \dfrac{K(2HM - M^2 - h_w^2)}{\lg \dfrac{R}{r_w}}$ |

Notes: $Q$ is the well seepage flux, $K$ is hydraulic conductivity, $H$ is the initial piezometric head, $s$ is the drawdown (i.e., dewatering depth), $r_w$ is the well radius, $M$ is the thickness of the aquifer, $R$ is the radius of influence of the pumping well, $h_w$ is the water level in the pumping well.

### 2.2. Laboratory Experiments

### 2.2.1. Seepage Sand-Tank and Boundary Conditions

Laboratory sand-tank axisymmetric radial water flow experiments were conducted to investigate the seepage to the ARW in an unconfined aquifer. The outer boundary head was at the elevation of 60 cm and kept constant for the entire test. The water entering the ARW was discharged freely under gravity, and the piezometric head inside the ARW was zero. The boundary conditions are shown Figure 2. The sand-tank model is generally a downscaled model of the aquifer [55].

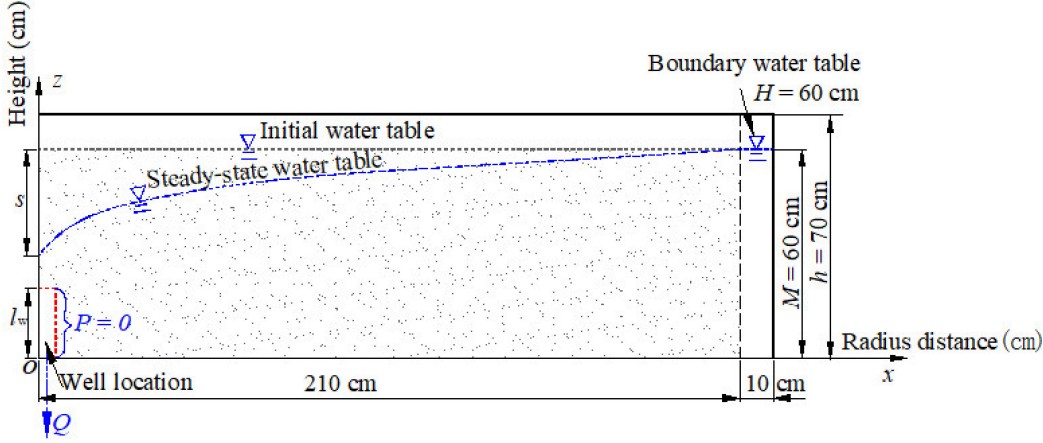

**Figure 2.** Schematic depicting initial and boundary conditions of the radial flow to a partially penetrating well in the sand-tank tests.

The tank was fabricated from steel plates and acrylic sheets, representing a 60° sector, i.e., a 1/6 portion of a circular flow system for saving experimental materials, with a radius of 210 cm and height of 70 cm. A design schematic and a photograph of the sand-tank assembly are shown in Figure 3. The outer boundary water head was maintained at the elevation of 60 cm above the bottom of the sand-tank, and water was supplied into the sand-tank from a plastic cistern. The outflow from the sand-tank was monitored by an electronic scale. The overflowing water and the well discharge were collected in separate water collection vessels for cyclic utilization. To measure the hydraulic pressure,

96 piezometers with water-level monitoring tubes were installed in six layers at different elevations and distances from the modeled ARW well. The inside diameter of the piezometers was 4 mm, shown in Figure A1 (see Appendix A). The piezometer storage effect is irrelevant because steady state water levels were measured. Water levels in the piezometer tubes were recorded using a camera, which is shown in Figure 3a.

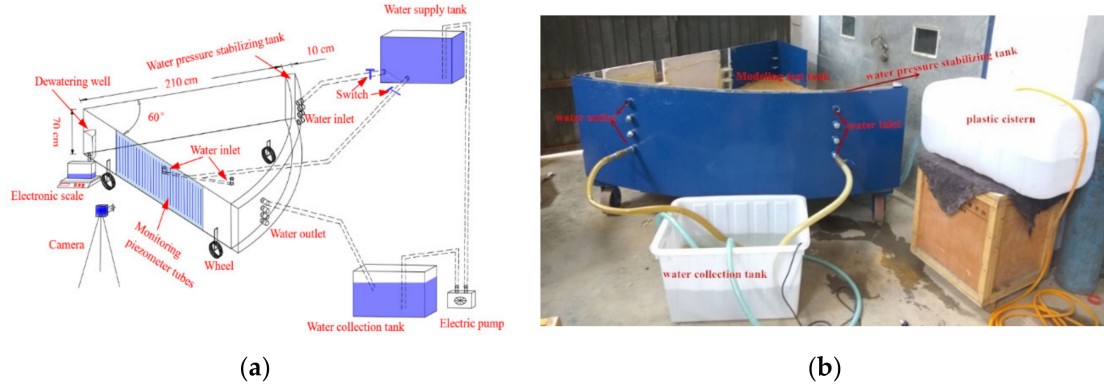

(**a**)                                                             (**b**)

**Figure 3.** Schematic design diagram (**a**) and a photograph (**b**) of the seepage sand-tank used for laboratory experiments.

### 2.2.2. Ascending Relief Well (ARW) Design

Laboratory tests were conducted to investigate the performance of 24 types of ARWs with different radii—4.5 cm, 9.5 cm, 14.5 cm and 19.5 cm, and lengths—2 cm, 5 cm, 10 cm, 20 cm, 30 cm and 60 cm, corresponding to the degrees of penetration 1/30, 1/12, 1/6, 1/3, 1/2 and 1, as shown in Figure A2. To prevent migration of sand particles into the testing wells, 1-mm radius holes with 4 mm pitches were made in the well screens and tops, and the well screens were wrapped up by stainless steel filters with 0.25 mm diameter openings and 0.18 mm diameter steel wire.

### 2.2.3. Laboratory Testing Protocol

#### Loading the Sand-Tank

The tank was filled up with preliminary washed river coarse sand, the particle size distribution curve of a sand sample used for the sand-tank experiment is shown in Figure A3. To preclude possible seepage along the sand-tank wall, a thin layer of vaseline was first besmeared along the inner surface of the tank wall. Six 10 cm thick sand layers were loaded into the tank layer by layer (in total 1.38 m$^3$), and 16 piezometers were installed between each two layers–in all 96 piezometers.

#### Sand Saturation

To saturate the sand, water was injected through two bottom inlet ports, while the water supply level was set at the 60 cm elevation above the tank bottom. During the tank saturation, the rising water level in the tank allowed for air escaping upward from the sand, which reduced the volume of entrapped air [56]. When the water table reached the elevation of 60 cm, water injection was continued for about two hours to allow for expelling the remaining entrapped air from the sand. Then, the bottom water inlets were shut off and kept closed during the dewatering process, and the water was diverted from the plastic cistern to the water pressure-stabilizing tank to maintain the stable upper boundary water level. The estimated saturated hydraulic conductivity of sand was about 0.6 cm/s. During dewatering, to assess the flux, water samples were collected and weighed, simultaneously with camera recording and measurements of piezometric heads in monitoring tubes.

### 2.3. Numerical Simulations

Numerical simulations of flow in the aquifer and the seepage from the ARW were carried out using a software package, MIDAS GTS NX. This software package is based on finite-element simulations for geotechnical design applications, such as deep foundations, excavations, complex tunnel systems, seepage, consolidation analysis, embankment design, dynamic and slope stability analysis—see http://en.midasuser.com/product/gtsnx_overview.asp. Two types of 3-D numerical simulations were conducted: one was to replicate the sand-tank laboratory experiment (Figure A4a), and the other one was to simulate the extended flow field domain (Figure A4b).

The 3-D extended modeling domain was 1200 m by 1200 m in the plan and 120 m in the vertical direction, with a 60 m homogeneous aquifer and a 60 m aquiclude overlying the aquifer. The ARW was located at the center of the domain, shown in Figure A4b. Simulations were performed to assess water flow to a single partially ARW and a fully penetrating ARW with different radii (0.1 m, 0.2 m, 0.5 m, 1 m, 2 m, 4 m, 6 m and 10 m), the well length (from 1 to 60 m with the 1 interval), and the outer boundary heads (60 m, 70 m, 90 m, 120 m, 150 m, 180 m, 210 m, and 240 m). To simulate the discharge from the ARW under gravity, the piezometric head inside the ARW was set to zero. The aquifer saturated hydraulic conductivity was 0.4 m/d. The fixed water pressure head (Dirichlet-type) boundary conditions were assigned at the outer boundary of the flow domain and the ARW well screen.

## 3. Results of Laboratory Sand-Tank Tests

To present the results of investigations, all distances and hydraulic heads are given as dimensionless values being normalized by the aquifer thickness $M$ for the confined aquifer or the original water level, $H$, above the unconfined aquifer bottom. For example, for the confined aquifer: $R_{nor} = \frac{R_x}{M}$, $H_{nor} = \frac{H}{M}$, $p_{nor} = \frac{p}{M}$, $l_{nor} = \frac{l_w}{M}$, $s_{nor} = \frac{s}{M}$ and $h_{nor} = \frac{h}{M}$, where $R_x$ is the radial distance, $H$ is the outer boundary water level, $p$ is the piezometric head, $l_w$ is the ARW penetrating length into the aquifer, $s$ is the drawdown, and $h$ is the height above the aquifer bottom. Here, $l_{nor}$ is also the ARW degree of penetration into the aquifer. The well radius $r_w$ was normalized as $r_{nor} = \frac{r_w}{r_1}$, using $r_1 = 1$ cm for the lab experiment, and 1 m for numerical modeling. The seepage fluxes $Q_{r1}$ for the fully penetrating well of radius $r_1 = 1$ cm (for lab experiment) and 1 m (numerical modeling), which were calculated for the case of complete drawdown, were used to determine the dimensionless seepage fluxes given by $Q_{nor} = \frac{Q}{Q_{r1}}$. The results are given below using dimensionless parameters.

### 3.1. Piezometric Head Distribution

Figure 4 demonstrates the piezometric head distribution vs. the distance from the well in the sand tank experiments for the case of the large diameter fully penetrating well, with $r_{nor} = 19.5$ and $l_{nor} = 1$. For large values of the well radius and length, the piezometric heads were zero at different heights above the ARW center with a maximum seepage flux. The seepage characteristics in the vicinity and at the ARW are practically the same as those in the case of a fully penetrating pumping well. Figure 5 shows the piezometric head distribution vs. distance from a partially penetrating ARW with a small degree of penetration of $l_{nor} = 1/30$ and a radius of $r_{nor} = 4.5$. The figure shows the zero piezometric head inside and at the top of the ARW, and the piezometric heads above the well top being greater than zero, indicating the downward flow. Figure 6 depicts the piezometric head distribution along the height of the sand-tank at the well center $R_{nor} = 0$ and at the radial distance $R_{nor} = 1/6$. In particular, the piezometric heads at the heights of $h_{nor} = 1/6$ and $h_{nor} = 1/3$ above the ARW center were nearly the same.

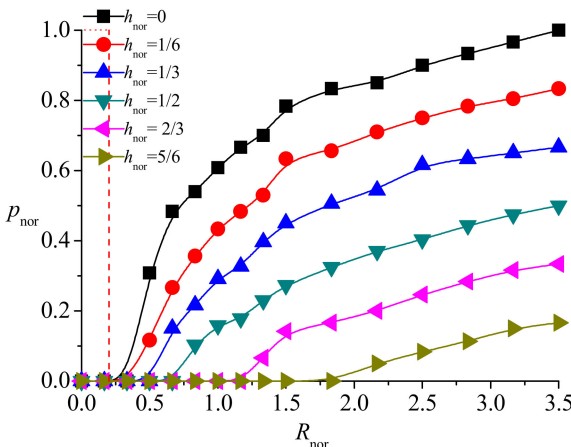

**Figure 4.** Piezometric head $p_{nor}$ distribution for the case of $r_{nor}$ = 19.5 and $l_{nor}$ = 1, i.e., a fully penetrating well in the laboratory sand-tank experiments.

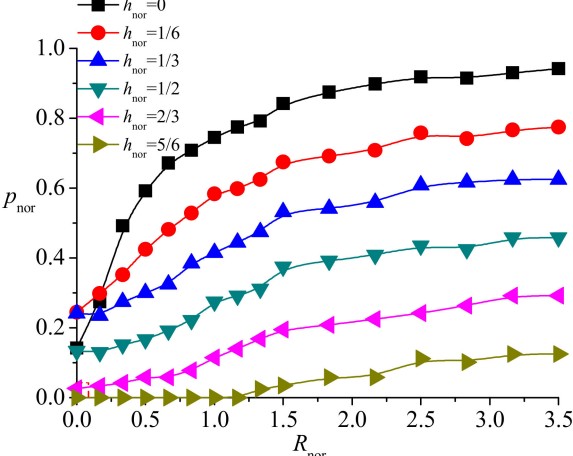

**Figure 5.** Piezometric head $p_{nor}$ distributions for the case of $r_{nor}$ = 4.5 and $l_{nor}$ = 1/30 in the laboratory sand-tank experiments.

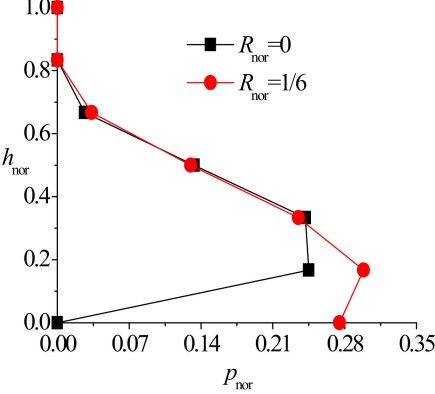

**Figure 6.** Piezometric head $p_{nor}$ distribution at radial distances $R_{nor}$ of 0 and 1/6 for the case of $r_{nor}$ = 4.5 and $l_{nor}$ = 1/30 in the laboratory experiment.

Figure 7 shows the 2-D plots of the piezometric head distribution (for the case of $r_{nor}$ = 4.5) for different values of the ARW degree of penetration. The experimental results show that the piezometric head inside the ARW was zero, and a piezometric head dropped in the vicinity of the partially penetrating ARW. The water table is determined at the elevation where the piezometric heads is zero.

Figures 5 and 6 show that the piezometric heads in the vicinity and above the partially penetrating ARW were greater in the middle and smaller at the top and bottom of the well. With the increase of the radial distance from the ARW, this effect gradually disappeared, and the piezometric heads at different depths increased with depth (see Figures 4 and 5).

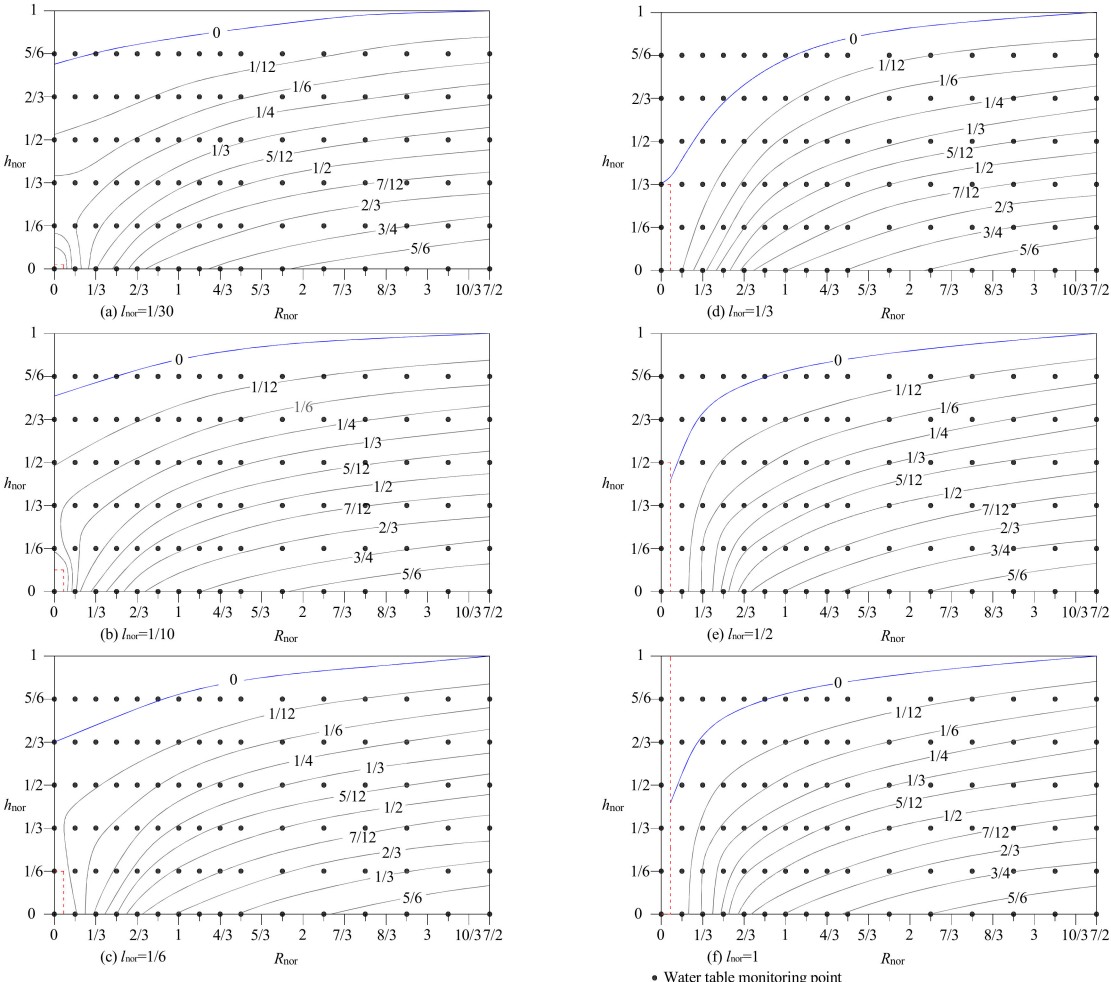

**Figure 7.** Piezometric head $p_{nor}$ distribution for the case of $r_{nor} = 4.5$ for different penetrating lengths $l_{nor}$. For partially penetrating wells: (**a**) $l_{nor} = 1/30$, (**b**) $l_{nor} = 1/12$, (**c**) $l_{nor} = 1/6$, (**d**) $l_{nor} = 1/3$, (**e**) $l_{nor} = 1/2$, and for a fully penetrating well: (**f**) $l_{nor} = 1$. The dashed red lines indicate the length of the well screened interval.

The effect of the well length on the water head distribution can be described as follows. For $l_{nor} \geq 1/2$, the piezometric head above the well is zero. For $l_{nor} \leq 1/3$, the piezometric head above the well is greater than zero. Because experiments were conducted for only six values of ARW length for $r_{nor} = 4.5$, it can be deduced that the critical length of the ARW penetration $l_{nor}$ ranges from 1/3 to 1/2.

The critical degree of ARW penetration of a confined aquifer $\zeta_c = l_c / M$ is defined as the ratio of the well critical length to the aquifer thickness $M$. For an unconfined aquifer, $\zeta_c = l_c / H$, where $H$ is the original water level above the unconfined aquifer bottom.

The influence of the well radius, length and boundary aquifer head on the critical degree of penetration of an ARW will be evaluated using the results of numerical modeling summarized below in Section "Numerical simulation results."

### 3.2. Seepage Flux

Figure 8 demonstrates that the seepage flux ($Q_{nor}$) increases asymptotically with the increase of the degree of penetration $l_{nor}$, which can be described by an exponential relationship (with $R^2 > 0.96$) given by:

$$\begin{cases} Q_{nor} = Q_{nor}^0 + A_0 \times \exp(B_0 \times l_{nor}), & \text{for } l_{nor} < \zeta_c \\ Q_{nor} = Q_{nor}^1 + A_1 \times \exp(B_1 \times \zeta_c), & \text{for } l_{nor} \geq \zeta_c \end{cases} \tag{1}$$

where $Q_{nor}^0$, $A_0$ and $B_0$, $Q_{nor}^1$, $A_1$ and $B_1$ are fitting parameters, and $\zeta_c$ is the critical degree of penetration.

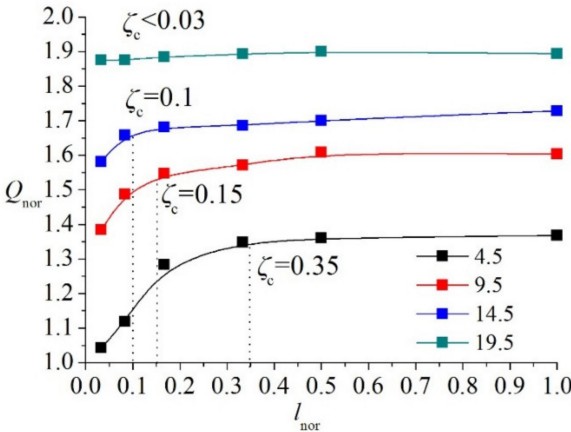

**Figure 8.** Relationship between the seepage flux $Q_{nor}$ and the degree of penetration $l_{nor}$ for different well radii.

Figure 8 demonstrates that for the small well radii, the seepage flux increased initially rapidly, and then practically stabilized for $\zeta_c > 0.35$. When the well radius is large, the seepage flux reaches the maximum value at a relatively small degree of penetration. The critical degree of penetration $\zeta_c$ decreases with the increase in the well radius, such as $\zeta_c = 0.1$ for $r_{nor} = 14.5$, $\zeta_c = 0.15$ for $r_{nor} = 9.5$, and $\zeta_c = 0.35$ for $r_{nor} = 4.5$. (The dependence of the critical degree of penetration $\zeta_c$ on the well radius and the boundary head, based on the results of numerical simulations, is described below in Section "Numerical simulation results"). Figure 9 demonstrates a relationship between the seepage flux $Q_{nor}$ and the well radius $r_{nor}$ given by

$$Q_{nor} = Q_{nor}^2 + A_2 \times r_{nor}^{B_2} \tag{2}$$

with $R^2 > 0.96$ and $B_2 < 1$, where $Q_{nor}^2$, $A_2$ and $B_2$ are fitting parameters, $r_{nor}$ is the well radius.

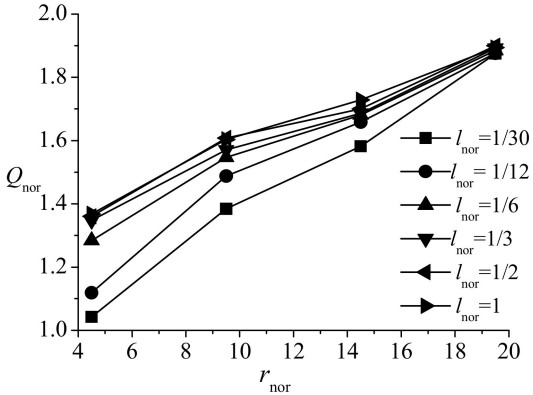

**Figure 9.** Relationship between seepage flux $Q_{nor}$ and well radius $r_{nor}$ for different degree of penetration $l_{nor}$, based on the results of the sand-tank experiments.

## 4. Numerical Simulation Results

### 4.1. Relationships between Seepage Flux, Well Length and Radius

The results of numerical simulations of the relationships between the seepage flux, well length and radius for different values of the boundary hydraulic head are shown in Figures 10 and 11. The relationship between the seepage flux and the degree of penetration can be described using an exponential function given by Equation (1) with $R^2 > 0.99$. The seepage flux increases with the increase in the well radius according to a power law relationship given by Equation (2) with $R^2 = 1$. The results of numerical simulations correspond to those from the laboratory experiments. Figures 10 and 11 also show that the seepage flux increases with the increase in the degree of penetration, and asymptotically reaches the maximum value for $l_{nor} \leq \zeta_c$. When the degree of penetration is greater than the critical value of $\zeta_c$, the seepage flux reaches the maximum value and remains practically constant as the degree of penetration continues to increase. The critical degree of penetration $\zeta_c$ decreases with the increase in the well radius, and increases with the increase of the boundary head.

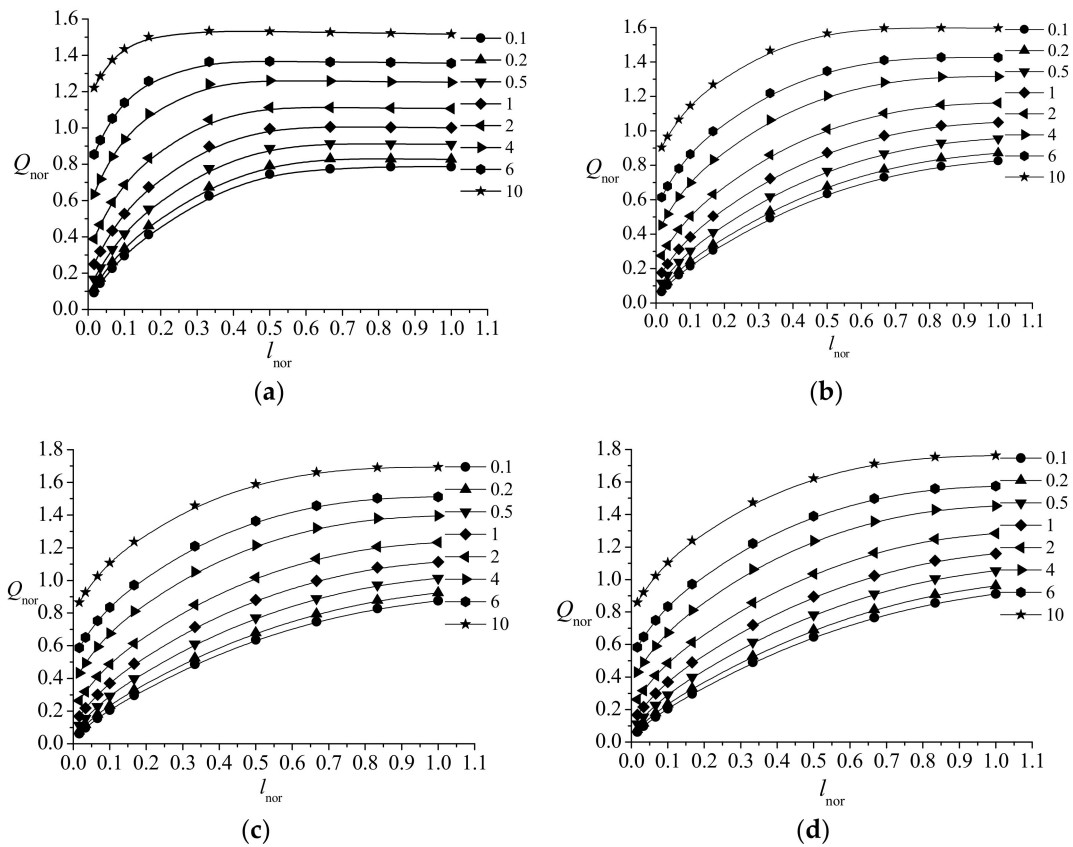

**Figure 10.** Relationships between the seepage flux $Q_{nor}$ and the degree of well penetration $l_{nor}$ for different boundary conditions: (**a**) $H_{nor} = 1$, (**b**) $H_{nor} = 2$, (**c**) $H_{nor} = 3$, and (**d**) $H_{nor} = 4$.

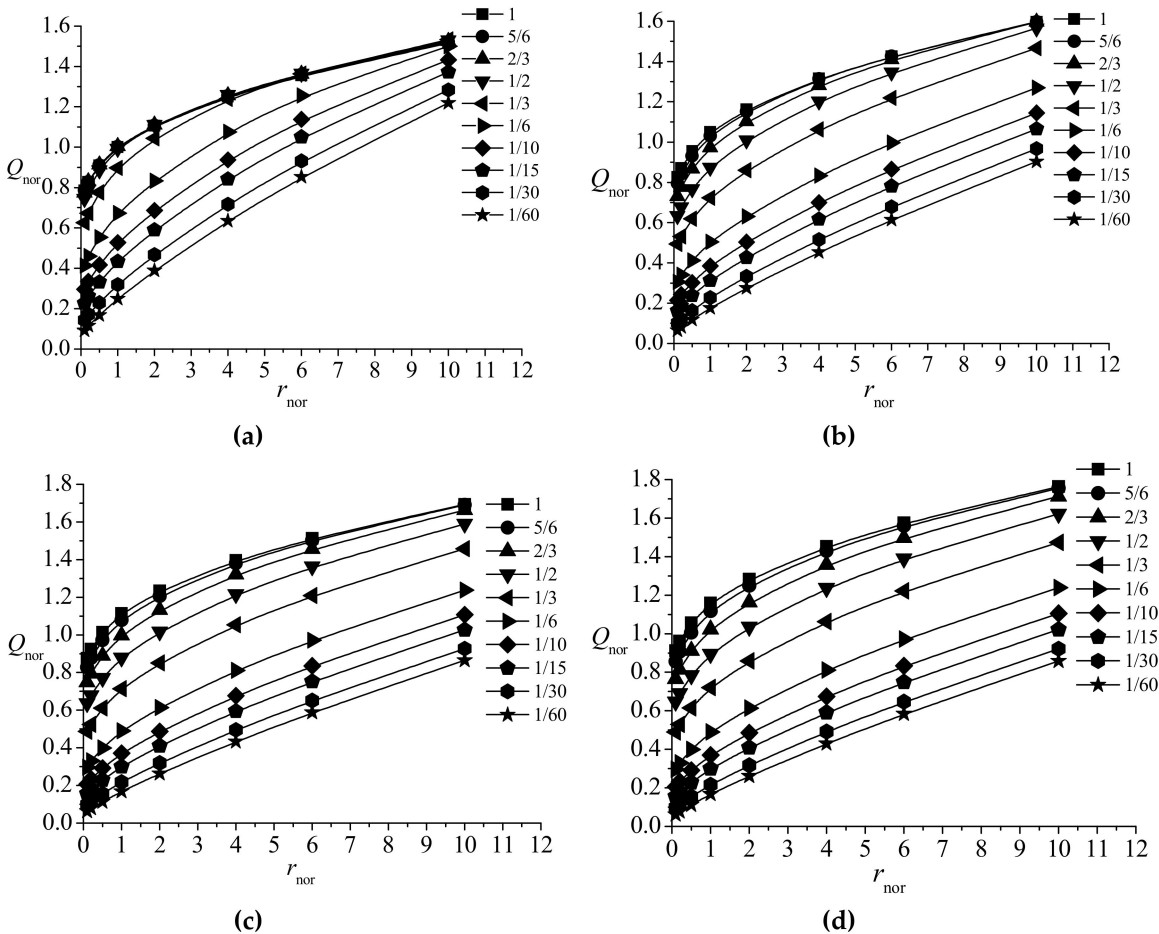

**Figure 11.** Relationships between the seepage flux $Q_{nor}$ and the well radius $r_{nor}$ for different boundary conditions: (**a**) $H_{nor} = 1$, (**b**) $H_{nor} = 2$, (**c**) $H_{nor} = 3$, (**d**) $H_{nor} = 4$.

*4.2. Seepage Characteristics*

The water piezometric head $p_{nor}$ distributions above the well for the case $H_{nor} = 1$ and $r_{nor} = 1.0$ in an unconfined aquifer are shown in Figure 12. For the partially penetrating ARW (for $l_{nor}$ from $1/60$ to $\frac{1}{2}$), shown in the subfigures (1)–(7) of Figure 12, the lines of the piezometric head of zero (i.e., indicating the water level) are located above the well screen (at $x = 0$), which correspond to the degree of penetration being less than the critical value of $\zeta_c$. For the partially penetrating wells (for $l_{nor}$ = 2/3 and 5/6), shown in subfigures (8) and (9) and a fully penetrating well, shown in subfigure (10) of Figure 12, the lines of the zero piezometric head are below the top of the well, which correspond to conditions exceeding the critical value of $\zeta_c$.

When the well length penetration exceeds the critical length $l_c$, the seepage flux of a partially penetrating ARW is the same as that of a fully penetrating well generating a maximum seepage flux. A similar pattern of the seepage flux was determined for a descending partially penetrating pumping well [57]. The piezometric head distribution characteristics determined using numerical modeling are consistent with the experimental results in the sand-tank described above in Section "Results of Laboratory Sand-Tank Tests."

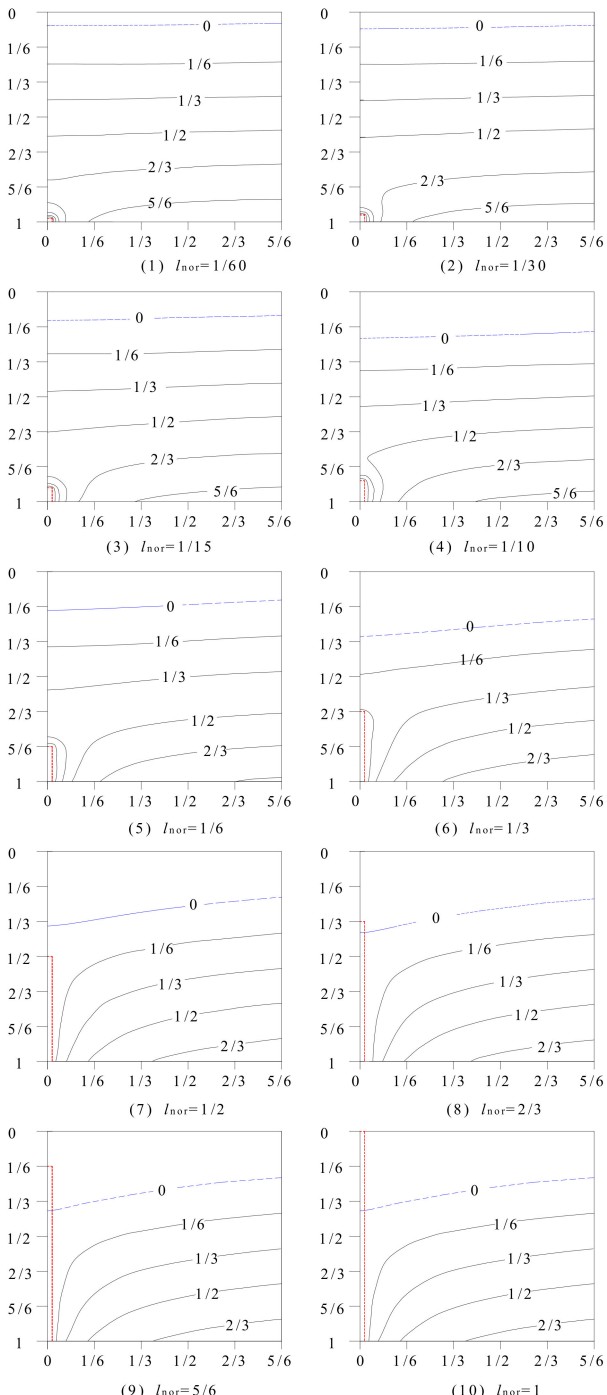

**Figure 12.** Results of simulations of the water piezometric head $p_{nor}$ distribution around the ARW in the unconfined aquifer for different well length $l_{nor}$: (**1**) $l_{nor}$ = 1/60, (**2**) $l_{nor}$ = 1/30, (**3**) $l_{nor}$ = 1/15, (**4**) $l_{nor}$ = 1/10, (**5**) $l_{nor}$ = 1/6, (**6**) $l_{nor}$ = 1/3, (**7**) $l_{nor}$ = 1/2, (**8**) $l_{nor}$ = 2/3, (**9**) $l_{nor}$ = 5/6, (**10**) $l_{nor}$ = 1 (for the case of $H_{nor}$ = 1 and $r_{nor}$ = 1.0). The *x*-axis is the normalized radial distance from the well.

Figure 13 depicts the flow net composed of streamlines and equipotential lines for the boundary head of $H_{nor}$ = 1 and $r_{nor}$ = 1.0. The flow net indicates that the equipotential lines above the ARW are concaved in the vicinity above the ARW. The curvature of the arched shape equipotential lines gradually increases with the well length increase, and when the equipotential line intersects the water table line, the equipotential line bifurcates into two lines, symmetrically distributed around the ARW. When the ARW's degree of penetration of exceeds the critical value of $\zeta_c$, the equipotential lines are

symmetrically distributed around the ARW. The streamlines shown in Figure 13 indicate that a single ARW can drain the entire aquifer regardless of the degree of penetration, and even for the degree of the penetration less than $\zeta_c$, groundwater flow from the entire aquifer thickness is directed into the ARW.

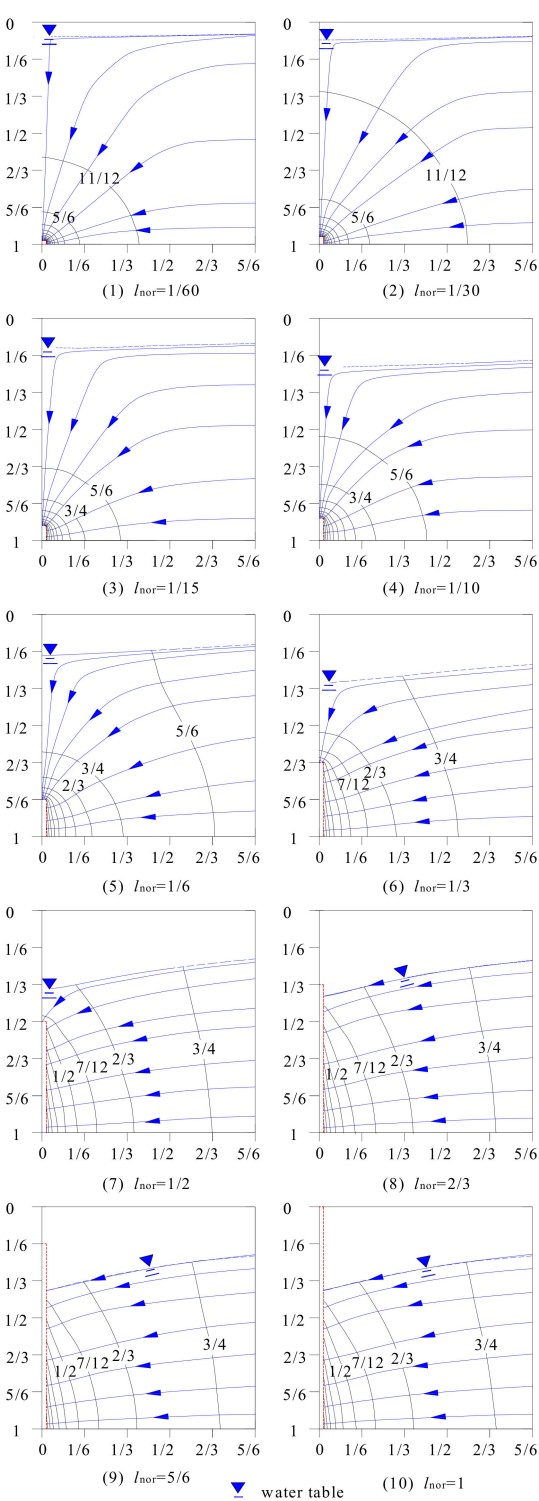

**Figure 13.** The flow net around the ARW in the unconfined aquifer for different well length $l_{nor}$: (**1**) $l_{nor}$ = 1/60, (**2**) $l_{nor}$ = 1/30, (**3**) $l_{nor}$ = 1/15, (**4**) $l_{nor}$ = 1/10, (**5**) $l_{nor}$ = 1/6, (**6**) $l_{nor}$ = 1/3, (**7**) $l_{nor}$ = 1/2, (**8**) $l_{nor}$ = 2/3, (**9**) $l_{nor}$ = 5/6, (**10**) $l_{nor}$ = 1 (for the case $H_{nor}$ = 1 and $r$ = 1.0). The *x*-axis is the radial distance from the well.

### 4.3. ARW's Critical Degree of Penetration

A summary of the calculations from the results of numerical simulation scenarios is presented in Table A2 and Figure 14. Figure 14 shows that $\zeta_c$ increases with $H_{nor}$ for different $r_{nor}$, according to the relationship given by:

$$\zeta_c = (-0.0488 \times r_{nor} + 0.98) + (0.0273 r_{nor} + 0.1118) \times \ln(H_{nor} + 0.0569 r_{nor} - 1.0214) \qquad (3)$$

where $r_{nor}$ is the ARW normalized well radius, $H_{nor} = H/M$ for the case of a confined aquifer, and $H_{nor} = H/H = 1$ for an unconfined aquifer.

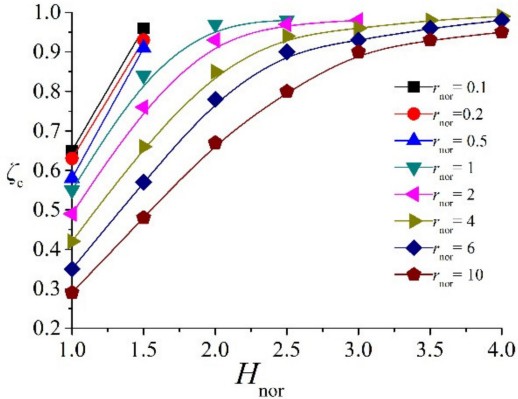

**Figure 14.** Relationship between $\zeta_c$ and $H_{nor}$ for different well radii.

For large $H_{nor}$ and a small well radius, for example, when $H_{nor} > 2.0$ and $r_{nor} < 0.2$, even for a fully penetrating well, the piezometric head above the well still exceeds zero, and the maximum seepage flux is not reached, as shown in Figure 14. Figure 15 presents a contour map of $\zeta_c$ for different $H_{nor}$ and $r_{nor}$ based on the results of numerical simulations. These results can be used for designing optimal ARWs to gain maximum seepage flux with minimum drilling lengths.

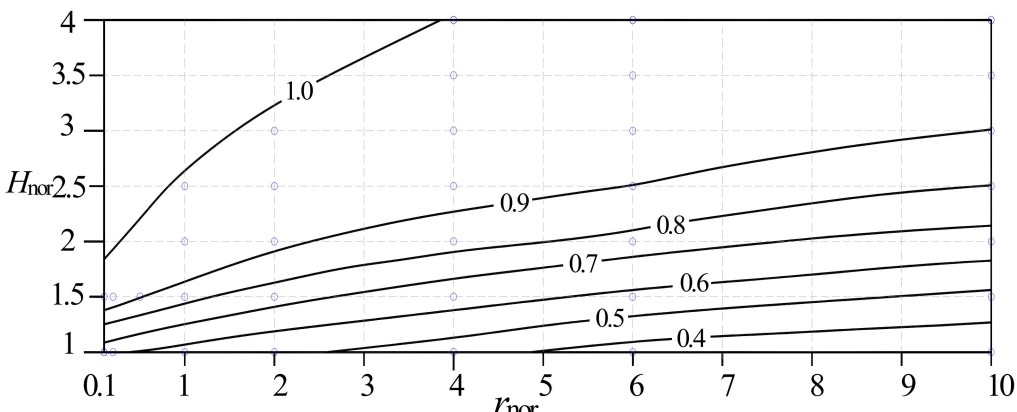

**Figure 15.** Type curves of $\zeta_c$ for as a function of $H_{nor}$ and $r_{nor}$.

### 4.4. Model Validation

Validation of the numerical model was conducted based on a comparison of the simulation results, performed using the code MIDAS GTS NX and calculations using the Dupuit formulae for a fully penetrating well in an unconfined aquifer, to the results of laboratory experiments, which are shown in Figure 16. The results of calculations of errors are listed in Table A1. Based on the results of calculations, the absolute relatively errors of the numerical simulations and Dupuit formulae calculations compared to the results of laboratory experiments are <7.65% and <4.49%, respectively.

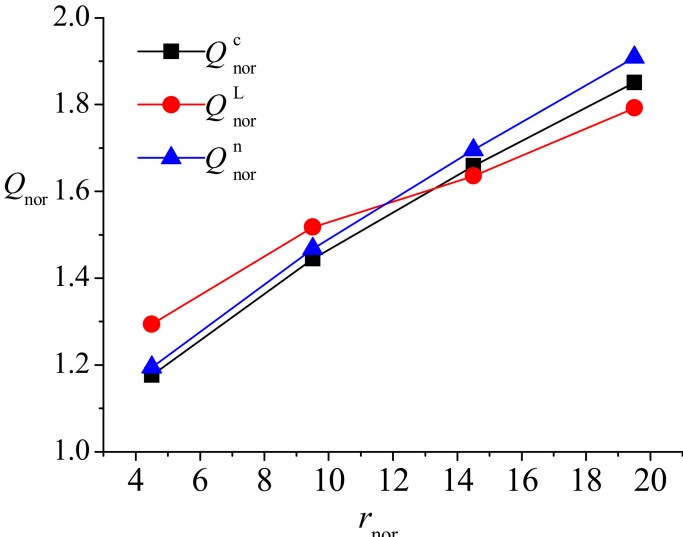

**Figure 16.** Comparison of the relationships between the seepage flux $Q_{nor}$ with $l_{nor} = 1$ and $r_{nor} = 4.5$, 9.5, 14.5 and 19.5 from numerical simulations, Dupuit formula and laboratory sand-tank experiments.

## 5. Modified Dupuit and Dupuit–Thiem Formulae for a Single ARW

### 5.1. Unconfined Aquifer

Table A3 presents the results of simulations of the seepage flux from numerical simulations ($Q_n$) and calculations using the Dupuit formula ($Q_s$) for a single partially penetrating ARW in an unconfined aquifer, shown in Figure 1a, for $r_{nor} = 1.0$ and the boundary head $H_{nor} = 1$. One can see that the seepage flux calculated using the Dupuit equation is smaller than that determined from numerical simulations. For $l_w < l_c$, the effective water level drawdown above the ARW can be determined as the value $s_{eff} = s + l_w$, and a modified version of the Dupuit equation for the seepage flux $Q_r$ can be given by:

$$Q_r = 1.366 \frac{K(2H - s_{eff})s_{eff}}{\lg \frac{R}{r_w}} = 1.366 \frac{K(2H - s_{eff})s_{eff}}{\lg \frac{10 s_{eff} \sqrt{K}}{r_w}} \tag{4}$$

with parameters identified on Figure 1a.

The seepage flux $Q_{nor}^r$ calculated using the modified Dupuit Equation (4) is also given in Table A3, and it is practically the same as that from numerical simulations, as shown in Figure 17a. Note that for $l_w \geq l_c$, the ARW seepage flux can be calculated from the Dupuit equation, as for a fully penetrating pumping well.

### 5.2. Confined Aquifer

For the case of a partially penetrating ARW in a confined aquifer, shown in Figure 1b, the seepage flux can be calculated using a modified version of the Dupuit–Thiem formula given by:

$$Q_r = 1.366 \frac{KM s_{eff}}{\lg \frac{R}{r_w}} = 1.366 \frac{KM s_{eff}}{\lg \frac{10 s_{eff} \sqrt{K}}{r_w}} \tag{5}$$

where the value of $s_{eff} = s + l_w$ is the effective piezometric head drawdown.

As an example, the results of calculations of the seepage flux for $r_{nor} = 1.0$ and the boundary head $H_{nor} = 2$ for different values of the well penetration are summarized in Table A4 and Figure 17b. Table A4 includes the seepage flux $Q_{nor}^n$ from numerical simulations, $Q_{nor}^s$ calculated using the Dupuit–Thiem formula, and $Q_{nor}^r$ from Equation (5). Table A4 also shows that the relative error of calculations $Q_{nor}^r$

using Equation (5) compared to the $Q^n_{nor}$ is from $-24.88\%$ to $1.1\%$, while the relative error of using the Dupuit–Thiem formula is from $-33.17\%$ to $-51.87\%$.

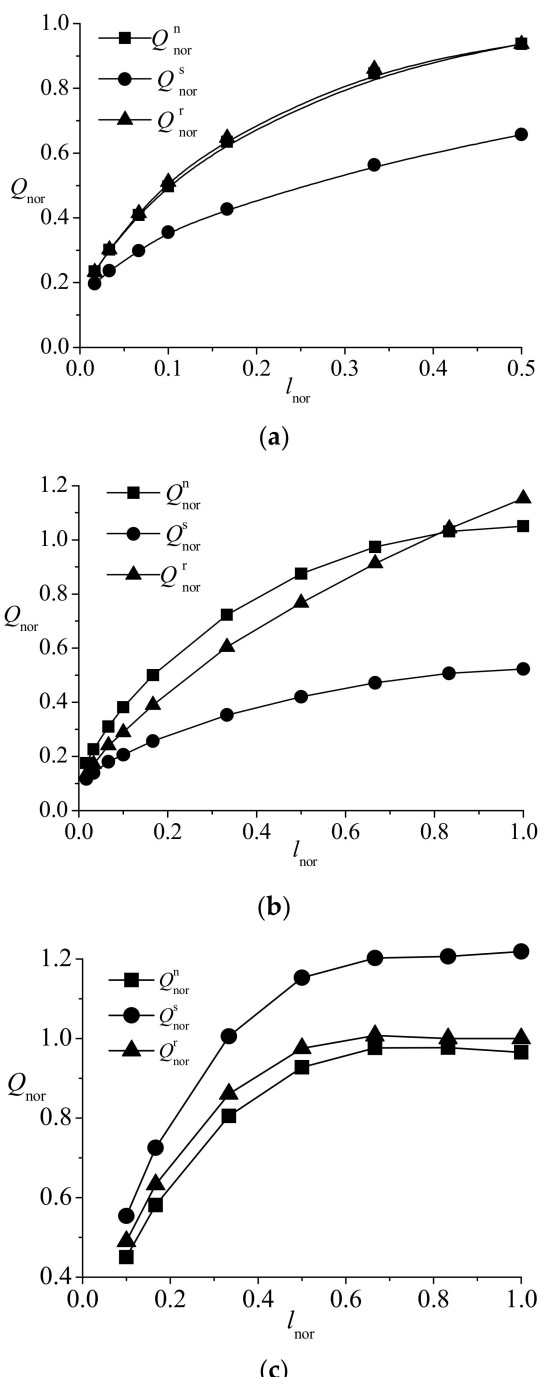

**Figure 17.** Seepage flux calculations using the results of numerical modeling ($Q_n$), original Dupuit ($Q_s$) and modified Dupuit formula ($Q_r$) for the cases of: (**a**) an unconfined aquifer, (**b**) confined aquifer, and (**c**) confined-unconfined aquifer.

### 5.3. Confined-Unconfined Aquifer

For the case of a partially penetrating ARW in a confined-unconfined aquifer, shown in Figure 1c, the results of simulations of the seepage flux can be described using a modified version of the Dupuit–Thiem formula given by:

$$Q_r = 1.366 \frac{K(2HM - M^2 - h_w{}^2)}{\lg \frac{R}{r_w}} = 1.366 \frac{K(2HM - M^2 - h_w{}^2)}{\lg \frac{10s_{eff} \sqrt{K}}{r_w}} \tag{6}$$

As an example, the results of numerical simulations, calculations using the Dupuit–Thiem formula, and formula (7), for $r_{nor} = 1.0$ and the boundary head $H_{nor} = 7/6$, are summarized in Table A5 and shown in Figure 17c. The relative error of calculations of $Q_r$ based on Equation (6), compared to numerical simulations, ranges from 2.37% to 8.87%.

### 5.4. Comparison of Modified Dupuit–Thiem Formulae with Results of Laboratory Sand-Tank Experiments

A comparison of calculations of the seepage flux based on the original and modified Dupuit formulae and sand-tank experiments are summarized in Figure 18 and Table A6. Figure 18 indicates that the seepage flux calculated using the modified Dupuit formula (Equation 6) is closer to the results of laboratory experiments. The seepage flux calculated from Dupuit formula is less than that from a modified Dupuit formula, with larger values being from laboratory experiments. However, the relative errors of $Q_{nor}^s$ compared to $Q_{nor}^r$ depend on the well length and the water table drawdown. As the well degree of penetration increases and the water table drawdown decreases, the relative error of $Q_{nor}^s$ compared to $Q_{nor}^r$ is increasing, as shown in Table A6. The data given in Tables A4–A6 confirm the validity of the application of a modified Dupuit–Thiem formula for calculations of the seepage flux for a single partially penetrating ARW.

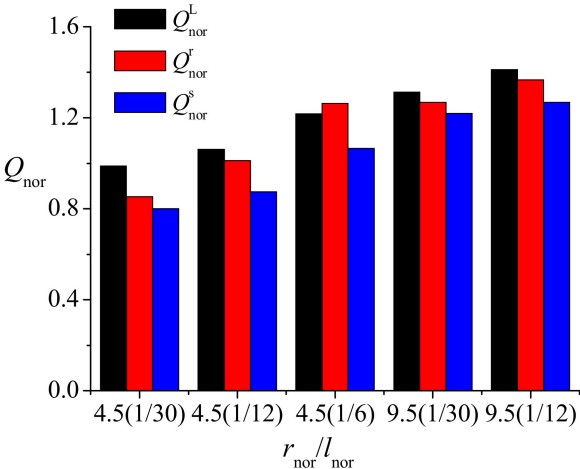

**Figure 18.** Comparison of the results of calculations of the seepage flux from the laboratory sand-tank experiments, original and modified Dupuit formulae.

## 6. Concluding Remarks

Laboratory sand-tank experiments and numerical simulations were conducted to assess the seepage characteristics of single partially and fully penetrating ARWs in homogeneous, isotropic unconfined, confined, and unconfined-confined aquifers. The seepage characteristics of the ARWs are affected by the seepage face boundary condition along an outflow boundary of the ARW, because the interior of the well casing remains open to the ambient air and is affected by the atmospheric pressure, so that water is discharging under gravity through the open bottom of the well casing. The experimental and modeling results indicate that (1) the relationships between the seepage flux and

the degree of penetration, and seepage flux and the well radius, can be described using the exponential function (Equation 1) and the power function (Equation 2), respectively; (2) for the ARW's degree of penetration less than the critical value $\zeta_c$, the piezometric head above the well is greater than zero with a piezometric head descent zone formed near the well.

The modeling results were used to develop (a) a series of type curves to evaluate the effect of the critical degree of penetration for different well radii and different aquifer water levels; and (b) the modified Dupuit and Dupuit–Thiem formulae by replacing the drawdown $s$ by an effective drawdown $s_{eff} = s+l_w$. It is shown that for the ARWs in an unconfined aquifer with the degree of penetration exceeding a critical value, the seepage flux can be calculated based on the original Dupuit formula, and for confined or unconfined-confined aquifers—based on the Dupuit–Thiem formulae for a fully penetrating well.

**Author Contributions:** Conceptualization, W.W., T.J., J.D., and B.F.; Data curation, W.W.; Formal analysis, W.W., B.F. and J.D.; Funding acquisition, W.W.; Experimental investigations, W.W. and Y.L.; Methodology, W.W., B.F. and T.J.; Project administration, W.W.; Resources, W.W. and Y.L.; Software, W.W. and T.J.; Supervision, B.F.; Validation, W.W., B.F., J.D. and Y.L.; Writing—original draft, W.W. and J.D.; Writing—review and editing, W.W., B.F., T.J. and J.D. All authors have read and agreed to the published version of the manuscript.

**Funding:** Wenxue Wang: 41602298; Wenxue Wang: 2020HYTP014; Boris Faybishenko: DE-AC02-05CH11231.

**Acknowledgments:** Wang, W.X. would like to acknowledge the support of the National Natural Science Foundation of China under Grant No. 41.602.298 and the Young Talent Promotion Project of Henan Province (2020HYTP014). B. Faybishenko was supported by the U.S. Department of Energy, Office of Science, Office of Advanced Scientific Computing under Contract No. DE-AC02-05CH11231.

**Conflicts of Interest:** The authors declare no conflict of interest.

## Appendix A

**Table A1.** Results of calculations of the errors of the seepage flux $Q_{nor}$ (for $l_{nor} = 1$ and $r_{nor} = 4.5$, 9.5, 14.5 and 19.5) from numerical simulations and Dupuit formula in comparison to the laboratory sand-tank experiments.

| Well Radius/$r_{nor}$ | Seepage Flux $Q_{nor}$ | | | Errors (%) | |
|---|---|---|---|---|---|
| | Lab Experiments | Dupuit Formula | Numerical Simulations | | |
| | $Q_{nor}^L$ | $Q_{nor}^s$ | $Q_{nor}^n$ | Errors of $Q_{nor}^n$ | Errors of $Q_{nor}^s$ |
| 4.5 | 1.294 | 1.16 | 1.195 | −7.65 | 3.02 |
| 9.5 | 1.517 | 1.425 | 1.467 | −3.30 | 2.95 |
| 14.5 | 1.636 | 1.636 | 1.696 | 3.67 | 3.67 |
| 19.5 | 1.792 | 1.826 | 1.908 | 6.47 | 4.49 |

**Table A2.** For different well radii and the boundary head.

| $r_{nor}$ \ $H_{nor}$ | 1 | 1.5 | 2.0 | 2.5 | 3.0 | 3.5 | 4.0 |
|---|---|---|---|---|---|---|---|
| 0.1 | 0.65 | 0.96 | | | | | |
| 0.2 | 0.63 | 0.93 | | | | | |
| 0.5 | 0.58 | 0.91 | | | | | |
| 1 | 0.55 | 0.84 | 0.97 | 0.98 | | | |
| 2 | 0.49 | 0.76 | 0.93 | 0.97 | 0.98 | | |
| 4 | 0.42 | 0.66 | 0.85 | 0.94 | 0.96 | 0.98 | 0.99 |
| 6 | 0.35 | 0.57 | 0.78 | 0.90 | 0.93 | 0.96 | 0.98 |
| 10 | 0.29 | 0.48 | 0.67 | 0.80 | 0.9 | 0.93 | 0.95 |

**Table A3.** Seepage flux $Q_{nor}$ calculations for an unconfined aquifer with $r_{nor} = 1.0$ and $H_{nor} = 1$.

| $l_{nor}$ | $s_{nor}$ | $s_{nor}+l_{nor}$ | Seepage Flux $Q_{nor}$ | | | Errors (%) | |
|---|---|---|---|---|---|---|---|
| | | | $Q_{nor}^n$ | $Q_{nor}^r$ | $Q_{nor}^s$ | Error of $Q_{nor}^r$ | Error of $Q_{nor}^s$ |
| 1/2 | 0.362 | 0.862 | 0.937 | 0.936 | 0.657 | −0.10 | −29.88 |
| 1/3 | 0.282 | 0.615 | 0.846 | 0.860 | 0.563 | 1.60 | −33.41 |
| 1/6 | 0.186 | 0.353 | 0.635 | 0.647 | 0.427 | 1.93 | −32.73 |
| 1/10 | 0.143 | 0.243 | 0.497 | 0.511 | 0.356 | 2.82 | −28.45 |
| 1/15 | 0.111 | 0.178 | 0.409 | 0.414 | 0.299 | 1.22 | −27.04 |
| 1/30 | 0.080 | 0.113 | 0.302 | 0.303 | 0.237 | 0.39 | −21.37 |
| 1/60 | 0.061 | 0.078 | 0.235 | 0.233 | 0.197 | −0.79 | −16.04 |

Notes: $Q_{nor}^s$ represents the results of calculations using the Dupuit formula, $Q_{nor}^n$ represents the results of numerical simulations, and $Q_{nor}^r$ represents the results of calculations using the modified Dupuit formula.

**Table A4.** Parameters used for calculations of the seepage flux for a confined aquifer with $r_{nor} = 1.0$ and $H_{nor} = 2$.

| $l_{nor}$ | $s_{nor}$ | $s_{nor}+l_{nor}$ | $Q_{nor}$ | | | Errors (%) | |
|---|---|---|---|---|---|---|---|
| | | | $Q_{nor}^n$ | $Q_{nor}^r$ | $Q_{nor}^s$ | Error of $Q_{nor}^r$ | Error of $Q_{nor}^s$ |
| 1 | 0.652 | 1.652 | 1.050 | 1.153 | 0.523 | 9.83 | −50.18 |
| 5/6 | 0.628 | 1.461 | 1.031 | 1.042 | 0.507 | 1.10 | −50.81 |
| 2/3 | 0.575 | 1.242 | 0.972 | 0.913 | 0.472 | −6.10 | −51.46 |
| 1/2 | 0.499 | 0.999 | 0.874 | 0.767 | 0.421 | −12.20 | −51.87 |
| 1/3 | 0.402 | 0.736 | 0.723 | 0.605 | 0.354 | −16.35 | −51.11 |
| 1/6 | 0.269 | 0.436 | 0.504 | 0.390 | 0.257 | −22.72 | −49.02 |
| 1/10 | 0.203 | 0.303 | 0.384 | 0.289 | 0.206 | −24.75 | −46.30 |
| 1/15 | 0.170 | 0.237 | 0.312 | 0.241 | 0.181 | −22.87 | −42.19 |
| 1/30 | 0.120 | 0.153 | 0.227 | 0.172 | 0.139 | −24.47 | −38.88 |
| 1/60 | 0.096 | 0.112 | 0.176 | 0.132 | 0.118 | −24.88 | −33.17 |

**Table A5.** Parameters used for calculations of the seepage flux for a confined-unconfined aquifer with $r_{nor} = 1.0$ and $H_{nor} = 7/6$.

| $l_{nor}$ | $s_{nor}$ | $h_{nor}^w$ | $s_{nor}+l_{nor}$ | $Q_{nor}$ | | | Errors (%) | |
|---|---|---|---|---|---|---|---|---|
| | | | | $Q_{nor}^n$ | $Q_{nor}^r$ | $Q_{nor}^s$ | Error of $Q_{nor}^r$ | Error of $Q_{nor}^s$ |
| 1 | 0.391 | 0.000 | 0.779 | 0.965 | 1.000 | 1.219 | 3.64 | 26.28 |
| 5/6 | 0.411 | 0.000 | 0.789 | 0.977 | 1.000 | 1.207 | 2.37 | 23.50 |
| 2/3 | 0.405 | 0.095 | 0.786 | 0.977 | 1.007 | 1.202 | 3.16 | 23.09 |
| 1/2 | 0.350 | 0.317 | 0.758 | 0.928 | 0.976 | 1.153 | 5.16 | 24.25 |
| 1/3 | 0.279 | 0.555 | 0.723 | 0.805 | 0.860 | 1.006 | 6.84 | 24.88 |
| 1/6 | 0.192 | 0.808 | 0.679 | 0.581 | 0.633 | 0.725 | 8.87 | 24.73 |
| 1/10 | 0.147 | 0.920 | 0.657 | 0.451 | 0.490 | 0.554 | 8.82 | 22.90 |

**Table A6.** Comparison of seepage flux calculations using data from the laboratory experiments, original and modified Dupuit formulae. Errors of original and modified Dupuit formulae are calculated in comparison to the results of laboratory sand-tank experiments.

| $r_{nor}(l_{nor})$. | $s_{nor}$ | $Q_{nor}$ | | | Errors (%) | |
|---|---|---|---|---|---|---|
| | | Lab Experiments | Modified Dupuit Formula | Dupuit Formula | | |
| | | $Q_{nor}^{L}$ | $Q_{nor}^{r}$ | $Q_{nor}^{s}$ | Errors of $Q_{nor}^{r}$ | Errors of $Q_{nor}^{s}$ |
| 4.5(1/30) | 0.20 | 0.988 | 0.853 | 0.801 | −13.66 | −18.93 |
| 4.5(1/12) | 0.23 | 1.061 | 1.011 | 0.875 | −4.71 | −17.53 |
| 4.5(1/6) | 0.33 | 1.217 | 1.264 | 1.066 | 3.86 | −12.41 |
| 9.5(1/30) | 0.37 | 1.313 | 1.267 | 1.220 | −3.50 | −7.08 |
| 9.5(1/12) | 0.40 | 1.411 | 1.366 | 1.267 | −3.19 | −10.21 |

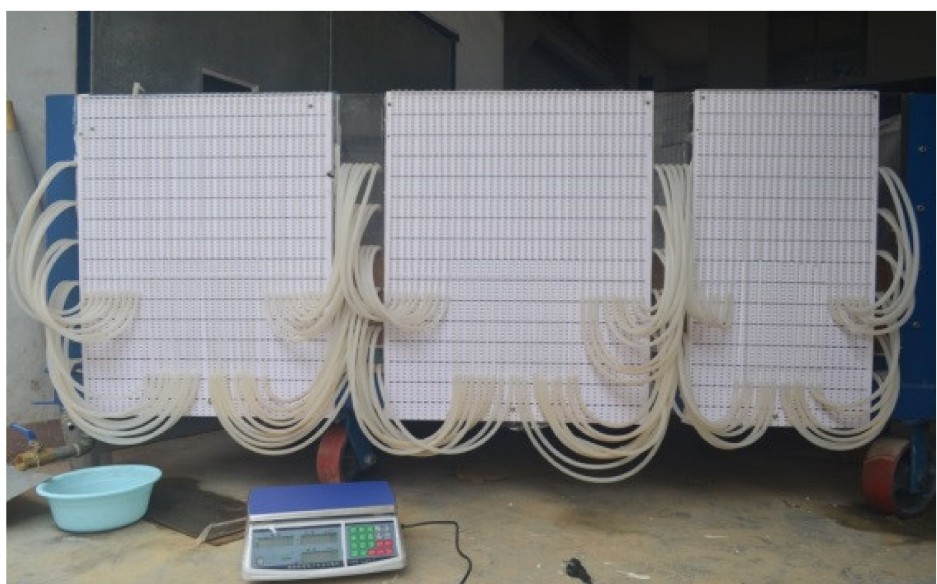

**Figure A1.** Photograph of the monitoring piezometer tubes.

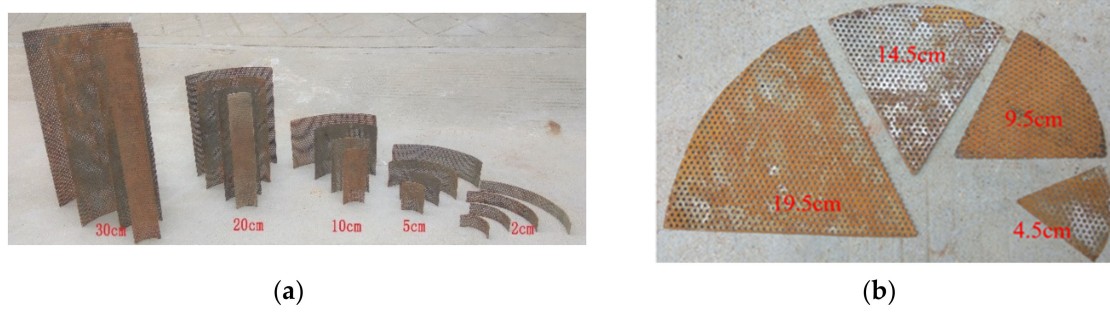

(**a**)      (**b**)

**Figure A2.** Photographs of seepage well screens (**a**) and well covers (**b**) used in the sand-tank experiments.

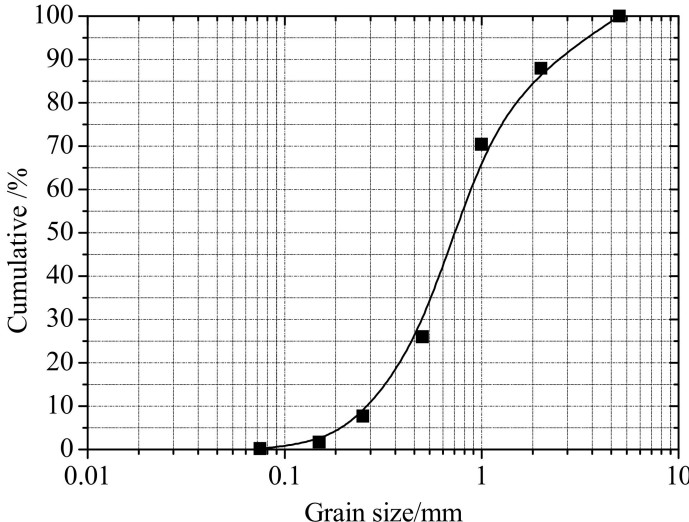

**Figure A3.** Particle size distribution curve of a sand used for the sand-tank experiments.

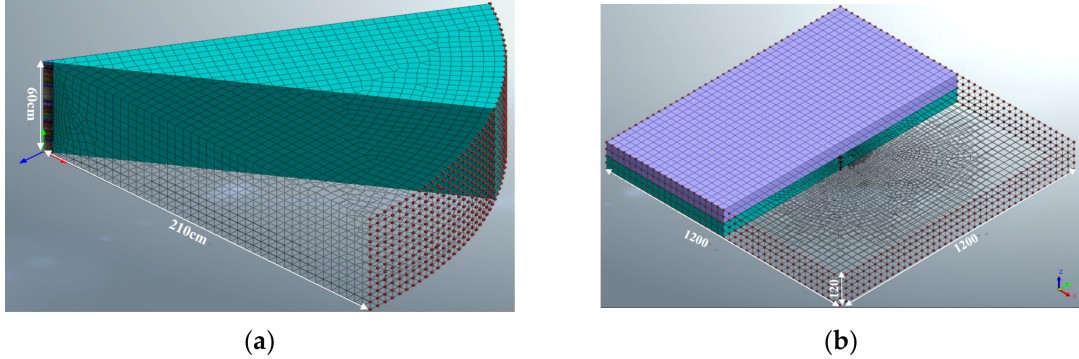

|   (a)   |   (b)   |

**Figure A4.** 3-D numerical discretization of the flow domains for simulations of: (**a**) the sand-tank experiments, and (**b**) the confined aquifer.

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
