# Peer review of "Seepage Characteristics of a Single Ascending Relief Well Dewatering an Overlying Aquifer"

_water, doi:10.3390/w12030919_

Round 1
Reviewer 1 Report
The manuscript is related to the description of a laboratory study to simulate, with a physical model, the drawdown caused by a well dewatering an aquifer with free drainage at the well bottom. The results are claimed to be compared with those of a numerical simulation model.
The topic and the proposed approach might be interesting in principle, but the manuscript needs to be improved from both the formal and the scientific points of view.
From the formal point of view, section 3 should present the results of the laboratory experiments, but figures 4, 5 and 6 refer to simulated values. Therefore, I was confused and I was not able to understand the workflow and the kind of represented results here and in the following. Therefore, it is necessary to revise the manuscript in order to provide a mre clear description of the results, otherwise it is impossible to assess the scientific merit of the work.
Lines 54-55. Figure 1 does not show the presence of an atmospheric boundary condition at the well bottom. Moreover, the case of figure 1.c should be considered more carefully, because it is necessary to permit air to enter in the originally saturated portion of the aquifer from the "impermeable" top surface.
Lines 63-64. The description of the Dupuit's assumption is not very accurate and rigorous. It does not refer to the well casing only.
Table 1. These formulas are not appropriate close to the well, where the horizontal flow approximation (hydraulic approximtaion for confined aquifers and Dupuit's approximation for unconfined aquifers) is not valid. They can be used far from the wells. These limitations should be discussed.
Lines 83-84. Here an head difference (whose dimensions are length) is compared with an hydraulic gradient (which is dimensionless). Please, explain your idea in a more clear and - above all - rigorous way.
Author Response
Comment 1.
From the formal point of view, section 3 should present the results of the laboratory experiments, but figures 4, 5 and 6 refer to simulated values. Therefore, I was confused and I was not able to understand the workflow and the kind of represented results here and in the following. Therefore, it is necessary to revise the manuscript in order to provide a more clear description of the results, otherwise it is impossible to assess the scientific merit of the work.
Responds: Figures 4, 5 and 6 are actually the results of the laboratory experiment. The figure captions have been revised to clearly indicate that these are the results of laboratory experiments.
Comment 2. Lines 54-55.
Figure 1 does not show the presence of an atmospheric boundary condition at the well bottom. Moreover, the case of figure 1.c should be considered more carefully, because it is necessary to permit air to enter in the originally saturated portion of the aquifer from the "impermeable" top surface.
Responds: The sentence “These diagrams show that the water discharge is taking place under gravity through the open well casing, indicating the atmospheric pressure boundary condition at the well-aquifer interface” is revised as follows: “These diagrams show that the water discharge is taking place under gravity through the open well casing.” The meaning of the statement of the atmospheric pressure boundary condition at the well-aquifer interface was explained on Lines 64-65. Note that the atmospheric boundary condition (i.e., the seepage face boundary condition) is formed at the inner surface area of the well casing, because the casing is atmospheric air (not water) saturated. Figure 1c demonstrates the formation of the atmospheric pressure boundary condition at the well-aquifer interface.
Comment 3. Lines 63-64.
The description of the Dupuit's assumption is not very accurate and rigorous. It does not refer to the well casing only.
Responds: We agree that the Dupuit's assumption “does not refer to the well casing only.” We revised the text and provided references to publications by Charnyi (1951) and Shercliff (1975), who showed that although the Dupuit formula is not accurate for calculations of the phreatic line close to the well, it provides accurate calculations of the seepage flux.
Comment 4. Table 1.
These formulas are not appropriate close to the well, where the horizontal flow approximation (hydraulic approximation for confined aquifers and Dupuit's approximation for unconfined aquifers) is not valid. They can be used far from the wells. These limitations should be discussed.
Responds: Seepage faces are formed along the well casing not only for pumping wells but also for ARWs. Actually, the Dupuit phreatic line is the same with the actual phreatic line when r>H, however, and it is lower than the actual phreatic line when r<H, because the seepage face is formed along the well casing. Charnyi (1951) showed that the Dupuit formula for an unconfined aquifer is valid not only for a hydraulic approximation, i.e., assuming that the groundwater velocity is independent of the height above the aquitard, but also for the rigorous hydrodynamic calculations. Although, the Dupuit formula is not accurate for calculations of the phreatic line for r<H, it provides accurate calculations of the seepage flux (Shercliff, 1975). Note that Shercliff (1975) paper concerns one- and two-dimensional models of steady seepage flow in unconfined aquifers and the relationship between them, and gives a new proof of Charnyi's result that one- and two-dimensional theory yield the same value for the flow rate in a horizontal aquifer or porous bed between vertical ends, and shows the extent to which it can be generalized to non-uniform or anisotropic media.
(References:
Charnyi, I.A, Strogoe dokazatel'stvo formuly Dyupyui dlya beznapornoi fil'tratsii s promezhutkom vysachivaniya, Doklady AN SSSR, 79(6): 937-940 (A rigorous derivation of Dupuit's formula for unconfined seepage with seepage surface, Doklady AN SSSR, 1951, 79(6): 937-940)
Shercliff, J.A., Seepage flow in unconfined aquifers, Journal of Fluid Mechanics, 71 (19), 1975: 181-192
Comment 5. Lines 83-84.
Here a head difference (whose dimensions are length) is compared with a hydraulic gradient (which is dimensionless). Please, explain your idea in a more clear and - above all - rigorous way.
Responds: The introduction was rewritten, and the sentence was deleted. The references to substantiate the atmospheric pressure boundary condition at the well-aquifer interface were added to the manuscript.
Reviewer 2 Report
REVIEW of the PAPER submitted by Wansg et al., to Water
GENERAL COMMENTS:
The objective of the paper “the assessment of groundwater drainage produced by an ascending Relief Well” is of potential interest for the Water readership. In my opinion, the manuscript needs to address/clarify some issues and some parts of the paper should be reorganized to imrpove the clarity of the exposition:
COMMENTS:
- INTRODUCTION:
The introduction should be improved. The authors should make an effort to clarify the novelty of the paper, from a methodological point of view and with respect to the tested case study. They do not state clearly what are the major contributions of this research, and the novelty and relevance of it. They should include a clearer definition of the knowledge gap, and them, to indicate the main novelties or differences of each of the proposed objectives with respect to the recent previous work developed by other authors about this topic, the case study and other pilots with similar conditions.
In the paper, they do not only compare the experiment with the numerical simulation. They also analyses the approximation obtained with some simple analytical solutions, modified version of Dupuit and Dupuit-Thiem solutions. Therefore, they should also specify it within the objective. In the literature we find several previous research papers in which classic numerical solutions are employed to test new analytical or simplified numerical solutions for different groundwater flow confined (See for example Llopis and Pulido-Velazquez, 2014, Liu et al., 2017) and unconfined problems (See for example Huang et al., 218, Pulido-Velazquez et al., 2006, 2007). Specifically, the modified version of Dupuit and Dupuit-Thiem solutions, that has been analyzed in many different research works (Knight, 2005).
These are only some examples, but they should add something in the line within the introduction. I will also move the formulation and Table 1 to Section 2, where the method is described.
In my opinion the assumption and limitations of the proposed method should be included and discussed more deeply in Section 2 (methodology) or even in the later ones, in which the results are presented and discussed. They should include examples in which similar assumptions are applied. The numerical solutions proposed could also consider the influence of the heterogeneity of the aquifer, which has been analyzed in many previous research works (Majumdar et al., 2008; Pulido-Velazquez et al., 2006). I would only mention some of the assumptions of the research work in the introduction if they help to clarify the novelty of the proposed analyses.
- METHODOLOGY: In my opinion a flowchart of the method should be included in a figure. It would make it easier to follow the different steps of the proposed methodology.
On the other hand, as I have already commented I will include the formations of Table 1 in this section.
Section 3, 4 and 5 (RESULTS). I will include not only the results but also the DISCUSSION of them. I would also specify it within the tittle. I miss a discussion in which more reference to the similarities and differences of the obtained results with other previous works. I will also include a limitation section where they should also identify and discuss (including example of other previous works) limitations/assumptions of the proposed method. They should for example discuss aspects as the heterogeneity of the hydrological impacts of climate change, including a brief analyses of potential explanatory variables. See for example the assessment at a regional scale in Continental Spain included in Pulido-Velazquez et al., 2018).
REFERENCES (The references included are only some examples among the many existing. I am not suggesting that the reviewer cite them)
Huang, Ching-Sheng & Yang, Tao & Yeh, Hund-Der. (2018). Review of Analytical Models to Stream Depletion Induced by Pumping: Guide to Model Selection. Journal of Hydrology. 561. 10.1016/j.jhydrol.2018.04.015.
Knight, John. (2005). Improving the Dupuit–Forchheimer groundwater free surface approximation. Advances in Water Resources. 28. 1048–1056. 10.1016/j.advwatres.2005.04.014.
Llopis-Albert, C. and Pulido-Velazquez, D., 2015. Using MODFLOW code to approach transient hydraulic head with a sharp-interface solution. Hydrol. Process 29, Issue 8, pages 2052–2064, 15. DOI: 10.1002/hyp.10354.
Liu, Xiao‐Xue & Shen, Shui-Long & Xu, Ye‐Shuang & Yin, Zhen‐Yu. (2017). Analytical approach for time‐dependent groundwater inflow into shield tunnel face in confined aquifer. International Journal for Numerical and Analytical Methods in Geomechanics. 42. 10.1002/nag.2760.
Majumdar, Pradeep & Sekhar, M. & Sridharan, K. & Mishra, G.. (2008). Numerical Simulation of Groundwater Flow with Gradually Increasing Heterogeneity due to Clogging. Journal of Irrigation and Drainage Engineering. 134. 400-404. 10.1061/(ASCE)0733-9437(2008)134:3(400).
Pulido-Velazquez D, Sahuquillo A, Andreu J, Pulido-Velazquez M., 2007a. A general methodology to simulate groundwater flow of unconfined aquifers with a reduced computational cost. Journal of Hydrology 2007, VOL 338 (1-2), 42-56, doi:10.1016/j.jhydrol.2007.02.009.
Pulido-Velazquez D, Sahuquillo A, Andreu J, 2006. A two-step explicit solution of the Boussinesq equation for efficient simulation of unconfined aquifers in conjunctive-use models. Water Resour Res 42 W05423, doi:10.1029/2005WR004473.
Pulido-Velazquez, D., Llopis-Albert, C., Peña-Haro, S., Pulido-Velazquez, M., 2011b. Efficient conceptual model for simulating the effect of aquifer heterogeneity on natural groundwater discharge to rivers. Advances in Water Resources 34 (2011) 1377–1389. doi:10.1016/j.advwatres.2011.07.010
Author Response
GENERAL COMMENTS:
The objective of the paper “the assessment of groundwater drainage produced by an ascending Relief Well” is of potential interest for the Water readership. In my opinion, the manuscript needs to address/clarify some issues and some parts of the paper should be reorganized to improve the clarity of the exposition:
- INTRODUCTION:
The introduction should be improved. The authors should make an effort to clarify the novelty of the paper, from a methodological point of view and with respect to the tested case study. They do not state clearly what are the major contributions of this research, and the novelty and relevance of it. They should include a clearer definition of the knowledge gap, and them, to indicate the main novelties or differences of each of the proposed objectives with respect to the recent previous work developed by other authors about this topic, the case study and other pilots with similar conditions.
In the paper, they do not only compare the experiment with the numerical simulation. They also analyses the approximation obtained with some simple analytical solutions, modified version of Dupuit and Dupuit-Thiem solutions. Therefore, they should also specify it within the objective. In the literature we find several previous research papers in which classic numerical solutions are employed to test new analytical or simplified numerical solutions for different groundwater flow confined (See for example Llopis and Pulido-Velazquez, 2014, Liu et al., 2017) and unconfined problems (See for example Huang et al., 218, Pulido-Velazquez et al., 2006, 2007). Specifically, the modified version of Dupuit and Dupuit-Thiem solutions, that has been analyzed in many different research works (Knight, 2005).
These are only some examples, but they should add something in the line within the introduction. I will also move the formulation and Table 1 to Section 2, where the method is described.
In my opinion the assumption and limitations of the proposed method should be included and discussed more deeply in Section 2 (methodology) or even in the later ones, in which the results are presented and discussed. They should include examples in which similar assumptions are applied. The numerical solutions proposed could also consider the influence of the heterogeneity of the aquifer, which has been analyzed in many previous research works (Majumdar et al., 2008; Pulido-Velazquez et al., 2006). I would only mention some of the assumptions of the research work in the introduction if they help to clarify the novelty of the proposed analyses.
Responds: According to the reviewer’s suggestions, the Introduction was rewritten and the objectives were clarified. Table 1 was moved to Section 2. A paragraph summarizing the underlying assumptions for analytical, experimental, and modeling studies, which were taken into consideration in this study, is added at the end Section 1.
We also cited in the paper the references recommended by the reviewer, and added several new references to a list of references of the manuscript. The authors appreciate the reviewer suggestion to include additional references.
The novelty of the paper is described in Sections 1 and 3, as well as in Conclusions. Publications on ARWs characteristics are limited, despite of the scientific and practical importance of research of ARWs. We believe this paper will provide a good reference to the scientists and engineers involved in the design and predictions of the aquifer dewatering projects using ARWs.
- METHODOLOGY: In my opinion a flowchart of the method should be included in a figure. It would make it easier to follow the different steps of the proposed methodology.
On the other hand, as I have already commented I will include the formations of Table 1 in this section.
Responds: The title of Section 2 was changed to Methods of Analytical, Experimental, and Modeling Investigations, and this section includes a description of analytical formulae, laboratory sand-tank experiments, and numerical simulations, which were used in this study. The original Dupuit and Dupuit-Thiem formulae were taken to present a theoretical basis, and were modified for calculations of the ARW seepage flux, based on the results of laboratory experiments and modeling. Numerical simulations were carried out to compare with the results of the laboratory experiments and to evaluate how boundary conditions may affect the seepage flux to the ARWs. Although we have not included the flowchart suggested by the reviewer, we modified the manuscript structure, which help easier follow the content of the paper. We moved Table 1 in Section 2.
- Sections 3, 4 and 5 (RESULTS).I will include not only the results but also the DISCUSSION of them. I would also specify it within the tittle. I miss a discussion in which more reference to the similarities and differences of the obtained results with other previous works. I will also include a limitation section where they should also identify and discuss (including example of other previous works) limitations/assumptions of the proposed method. They should for example discuss aspects as the heterogeneity of the hydrological impacts of climate change, including a brief analyses of potential explanatory variables. See for example the assessment at a regional scale in Continental Spain included in Pulido-Velazquez et al., 2018).
Responds: Section 3 includes both the results and detailed discussions of the results. A critical review of publications related to the similarities and differences of ARWs with pumping wells and tunnels is included in a revised manuscript. A paragraph including simplifying assumptions and limitations is added at the end of Section 1.
The topics of the effect of climate change and a regional scale research are not subjects of the current paper.
Section 4.4 of the revised manuscript includes a discussion of the numerical model validation. The validity of the results of calculations using modified Dupuit and Dupuit-Thiem formulae is given using a comparison with the results of laboratory sand-tank experiments in Section 5.4.
Reviewer 3 Report
The paper presents a laboratory investigation which is completed, clearly shown and described properly. Unfortunatelly in reviewer opinion results of laboratory study is not sufficient to be published in Water journal as a reaserch article. There is no clearly shown novelty; discussion with other scientists/previous publish papers is missed. Presenting a modified Dupuit formulae is not a good enough for scientific article. Beside of this, analyzing only isotropic sand-tank brings results of oversimplified environment of groundwater. In practice such aquifers are not observed so often. So the proposed modified formulae can't be use commonly in a practice. The next doubts is a scale effect. In my opinion experiments should be expand (larger tank with more complicated lithology).
Author Response
The paper presents a laboratory investigation which is completed, clearly shown and described properly. Unfortunatelly in reviewer opinion results of laboratory study is not sufficient to be published in Water journal as a reaserch article. There is no clearly shown novelty; discussion with other scientists/previous publish papers is missed. Presenting a modified Dupuit formulae is not a good enough for scientific article. Beside of this, analyzing only isotropic sand-tank brings results of oversimplified environment of groundwater. In practice such aquifers are not observed so often. So the proposed modified formulae can't be use commonly in a practice. The next doubts is a scale effect. In my opinion experiments should be expand (larger tank with more complicated lithology).
Responds: We have improved Section “Introduction” and clarified the novelty. We also cited additional references related to the topic of water inflow to tunnels. However, we respectfully disagree with the reviewer’s statement that presenting “modified Dupuit formulae is not a good enough for scientific article.” Review of scientific literature shows that despite of numerous publications over the last century on the evaluation of the validity of the Dupuit formulae, no modifications have been proposed to the analysis of ARWs. We believe that the developed modified formulae will be used by hydraulic engineers and scientists involved in the design and predictions of dewatering projects.
We agree that isotropic aquifers are not common in nature, but the evaluation of a scale effect was not a subject of this paper. The seepage characteristics of anisotropic and heterogeneous aquifers will be studied in a larger tank in the future. Nonetheless, we would like to refer to the paper by Shercliff (1975), who showed that the Dupuit formula can be generalized to non-uniform or anisotropic media (Shercliff, J.A., Seepage flow in unconfined aquifers, Journal of Fluid Mechanics, 71 (19), 1975: 181-192).
Round 2
Reviewer 1 Report
The manuscript has been strongly improved with the revision and it is now of acceptable quality for publication.
Author Response
Thanks a lot for your comments.
Comment: English language and style are fine/minor spell check required.
Response: Thank you for the comment—additional spell check will be provided.
Reviewer 2 Report
In my opinion, the reviewed performed has not properly addressed the comments that I pointed out. I miss more detailed reponses to my comments. Changes introduced in the manuscript in accordance with the cited comments should be included in the response specifying their location (line) within the new version of the paper. It would make it easier to follow if the comments have been considered and properly addressed. If there are some discrepancies between the reviewer comments and the authors’ opinión, they should try to justify their position. It is a little strange that the authors’ answers are significatively shorter than the comments introduced by the reviewer. I think that they should look them more carefully. For example, I miss a paragraph where the novelty of the paper is stated clearly in accordance with the knowledge gaps, which should be also clearly specified. I included some examples in my comments that could help to identify some missed points. I suggested to include a figure that summarise the methodology applied, but you not even mention anything about it in your answer. I continue missing a discussion in which more reference to the similarities and differences of the obtained results with other previous works. I would suggest that the authors provided detailed response to all my comments.
Author Response
We thankful to the reviewer for the comments and suggestions expressed in both rounds of the review of our manuscript. Although we are sorry to hear that the reviewer is not satisfied how we addressed reviewer’s comments, below are our answers to the reviewer’s 2nd review.
Comment: “In my opinion, the reviewed performed has not properly addressed the comments that I pointed out. I miss more detailed reponses to my comments. Changes introduced in the manuscript in accordance with the cited comments should be included in the response specifying their location (line) within the new version of the paper. It would make it easier to follow if the comments have been considered and properly addressed. If there are some discrepancies between the reviewer comments and the authors’ opinion, they should try to justify their position.”
Response: We have thoughtfully addressed the comments of all three reviewers, which helped us improve the manuscript, and we provided detailed responses. Because some of the three reviewers’ comments were alike, it is difficult to cite the lines that were changed according to specific comments of each of the reviewers. In our opinion, we provided reasonable answers to the reviewer’s comments, and also justified our position to comments that we disagree with.
Comment: “It is a little strange that the authors’ answers are significatively shorter than the comments introduced by the reviewer. I think that they should look them more carefully. For example, I miss a paragraph where the novelty of the paper is stated clearly in accordance with the knowledge gaps, which should be also clearly specified. I included some examples in my comments that could help to identify some missed points.”
Response: It is not uncommon that the authors’ answers are shorter than the reviewer’s comments. In the revised manuscript, according to the reviewer’s suggestions, the Introduction was rewritten and the objectives were clarified. A paragraph summarizing the underlying assumptions for analytical, experimental, and modeling studies, which were taken into consideration in this study, is added at the end Section 1. We also cited the references recommended by the reviewer, and added several new references to a list of references of the manuscript. The novelty of the paper is described in Sections 1 and 3, as well as in Conclusions.
Comment: “I suggested to include a figure that summarise the methodology applied, but you not even mention anything about it in your answer.”
Response: Looks like the reviewer has missed our answer given in the response to the 1st review: “Although we have not included the flowchart suggested by the reviewer, we modified the manuscript structure, which help easier follow the content of the paper.”
Comment: “I continue missing a discussion in which more reference to the similarities and differences of the obtained results with other previous works.
Response: In the revised manuscript we extended a review of scientific literature (included the references suggested by the reviewer), and showed that despite of numerous publications over the last century no similar experimental and modeling studies have been performed on the analysis of ascending relive wells. Therefore, we included in Section 4.4 of a revised paper the discussion on the numerical model validation, and in Section 5.4—the results of the validity of calculations using modified Dupuit and Dupuit-Thiem formulae using a comparison with the results of experimental laboratory sand-tank experiments.
Comment: “I would suggest that the authors provided detailed response to all my comments.”
Response: We believe that our answers to both the 1st and 2nd round of the review provide detailed answers to all reviewer comments.
Comment: English language and style are fine/minor spell check required.
Response: Thank you for the comment—additional spell check will be provided.
Reviewer 3 Report
After revision the paper is sufficient and I can recommend it to publish in Water.
Author Response
Thank you so much for your review on our manuscript.
Round 3
Reviewer 2 Report
In my opinion, you have not properly addressed the comments that I pointed out. It is a pity, because I consider that the papers is interesting, and it would not be very difficult for you to answer properly my questions. Although it would require to spend some time on it, I think that it might help you to clarify some part of the manucript. As I pointed out in my comments more detailed reponses to my comments should be provided. Changes introduced in the manuscript in accordance with the cited comments should be included in the response specifying their location (line) within the new version of the paper. It would make it easier to follow if the comments have been considered and properly addressed. If there are some discrepancies between the reviewer comments and the authors’ opinion, they should try to justify their position.